# Pressure dependence of the structural and optoelectronic properties of Pb-free perovskites LiSnX₃ (X = Br and Cl): A DFT approach

**Mohammed Noor S. Rammoo**  *, **Hameed T. Abdulla, Bewar M. Ahmad, Nawzad A. Abdulkareem**

Department of Physics, College of Science, University of Zakho, Duhok, Kurdistan Region, Iraq

\* mohammed.noor@uoz.edu.krd

**Data Availability Statement:** All relevant data are within the paper.

**Funding:** The author(s) received no specific funding for this work.

## Abstract

In this study, the structural, electronic and optical properties of cubic lead-free halide perovskites LiSnX₃ (X = Br and Cl) under hydrostatic pressure are investigated. The first-principle approach based on density functional theory (DFT) is employed. The exchange-correlation functional is treated using the generalized gradient approximation (GGA), specifically a variant of the Perdew–Burke–Ernzerhof (PBE) method. The aim of the study is to understand the effect of pressure on the properties of LiSnX₃ (X = Br and Cl), with a maximum pressure limit of 6 GPa. The results show a decreasing tendency in the energy band gap as pressure increases. In addition, a prominent reduction in the energy band gap is observed when the halogen atom is changed from Cl to Br under constant pressure. The calculations also investigate the density of states (DOS), showing variations in energy levels near the Fermi level under different pressures. For optical properties, density functional perturbation theory (DFPT) is used in conjunction with the Kramers-Kronig relation. Optical parameters such as the real and imaginary parts of the dielectric constant, refractive index, absorption coefficient, and wavelength are computed under different pressures to understand the optical response of the perovskites to the electromagnetic spectrum. The insights from this study highlight the fundamental properties of LiSnX₃ (X = Br and Cl) under different pressures, which could influence advancements in optoelectronic devices, photonic applications, and solar cell technologies. Moreover, this research contributes to the growing body of knowledge on lead-free halide perovskites, encouraging further developments in the field.

## 1. Introduction

The global drive for sustainable energy solutions has gained heightened importance, driven by concerns over depleting fossil fuel reserves and the escalating effect of climate change. Solar energy has emerged as a leading alternative, with photovoltaic technologies playing a fundamental role in harnessing solar power [1, 2]. Among these technologies, lead-based perovskite

**Competing interests:** The authors declare that they have no conflict of interest regarding the publication of this research.

solar cells (PSCs) have garnered substantial attention due to their potential to transform solar energy generation. Boasting exceptional power conversion efficiency and cost-effective production, PSCs are positioned as a disruptive innovation in the solar energy sector. The unique crystalline structure and superior optoelectronic properties of lead-based perovskite materials have elevated them to a prominent position in the photovoltaic market, challenging the long-standing dominance of silicon-based solar cells [3–5].

A recent study has reported that (PSCs) have achieved an efficiency of 25.8% [6], approaching the performance of conventional silicon solar cells, which stand at 26.6% [7]. Remarkably, the combination of silicon and perovskite in tandem solar cells has demonstrated even greater potential, with reported efficiencies as high as 29.8% [8]. However, despite these advancements, concerns regarding the environmental and health impacts of lead in PSCs have impeded their widespread adoption [9]. In response, researchers are increasingly focused on the development of lead-free perovskite materials, known as $ABX_3$. These materials offer a more sustainable and environmentally friendly alternative, addressing safety concerns while holding a significant promise for revolutionizing the PSC market and facilitating large-scale production. This shift has the potential to meet the growing request for cleaner, safer energy solutions, ultimately advancing the transition to more sustainable energy sources. Recent research by Ke and Kanatzidis (2019) has highlighted the exceptional properties of Sn-based perovskites in comparison to other lead-free alternatives. These perovskites demonstrate remarkable stability and exhibit superior performance, particularly in solar cell applications [10]. Significantly, studies have highlighted the promising optoelectronic properties of different $ASnX_3$ perovskites, including $CsSnX_3$ (X = I, Br, Cl) [11], $KSnCl_3$ [12], $TlSnF_3$ [13], $RbSnX_3$ (X = Cl, Br) [14, 15], $InSnCl_3$ [16], and $InSnX_3$ (X = I, Br, Cl) [17]. In addition, further computational investigations into Sn-based double perovskites, such as $Cs_2SnI_6$, have revealed their remarkable and distinct optoelectronic properties when compared to other lead-free perovskites [18]. These findings highlight the growing significance of Sn-based perovskites across various applications, particularly in solar energy and optoelectronics.

Recent studies on $TiSnF_3$ by Pingak [13] and Zaman et al. [19] have revealed an indirect band gap (R → M) with energy values of 0.75 eV and 0.63 eV, respectively. This indirect band gap in $TiSnF_3$ can be transformed into a direct band gap by substituting F with Cl, Br, or I. Recent research by Pingak et al. [17] demonstrated that replacing Cl with Br alters the band gap characteristics of these compounds. Furthermore, Singh's [20] theoretical investigation of $TlSnI_3$ in its orthorhombic structure identified Sn as a promising activator in $TiPbI_3$, suggesting its potential application in low-band-gap scintillators. Consequently, further exploration of lead-free perovskites, such as $TiSnCl_3$, $TlSnBr_3$, and $TiSnI_3$, is of great significance. Despite previous research efforts, there remains a notable gap in studies concerning Sn-based halide perovskites incorporating Li as the cation and X representing Cl, Br, or I. Jabar A. et al. [21] investigated the thermoelectric and optoelectronic properties of $LiSnX_3$ (X = Br or I). Under hydrostatic pressures up to 40 GPa, the structural, optoelectronic, and mechanical properties of non-toxic $CsSnCl_3$ metal halides have been studied through first-principles simulations [22, 23]. Recently, the mechanical, structural, magnetic, and optoelectronic properties of perovskite hydrides $XSnH_3$ (X = K and Li) have also been investigated [24]. More recently, a pressure-dependent DFT study of $MSnI_3$ (where M = K and Rb) was conducted to explore electronic phase transitions and enhance optoelectronic applications [25].

This study focuses on the investigation of the theoretical effects of hydrostatic pressure on the properties of $LiSnX_3$ (X = Br and Cl). Given the lack of experimental data on $LiSnX_3$ under such conditions, our work aims to address this gap. Using ab-initio calculations within the framework of (DFT), we systematically study how pressure influences the structural and optoelectronic properties of $LiSnX_3$ (X = Br and Cl). To ensure high accuracy, our calculations

use a dense 14 x 14 x 14 k-point mesh. Specifically, we investigate the band structure, total and partial densities of states (TDOS and PDOS), as well as the real and imaginary components of the dielectric function. In addition, we calculate the refractive index and absorption coefficient across different pressure regimes. This comprehensive theoretical study aims to provide valuable insights into how hydrostatic pressure affects the electronic and optical behavior of these cubic perovskites, showing their potential applications and behavior under varying conditions.

## 2. Computational method

The cubic perovskite structure of LiSnX₃ (where X = Br or Cl) adopts a unit cell belonging to the Pm–3m (#221) space group. Within this unit cell, five atoms are positioned at specific Wyckoff sites: Li occupies the 1b site (0.5, 0.5, 0.5), Sn resides at the 1a site (0.0, 0.0, 0.0), and the three X atoms are collectively positioned at the 3d site (0.0, 0.0, 0.5). Ab initio calculations were conducted using the plane-wave pseudopotential (PW-PP) method within the framework of (DFT), as implemented in the ABINIT package [26]. The Perdew-Burke-Ernzerhof (PBE) [27] (GGA) was selected to describe the exchange-correlation functional in the Kohn-Sham equations.

The Plane Wave Pseudopotential (PW-PP) method, using plane waves as the basis set for wave function expansion, was employed to model the valence-electron-ion interaction. This interaction was represented using norm-conserving, separable, dual-space pseudopotentials, specifically of the Gaussian type, as proposed by Goedecker, Teter, and Hutter (GTH) [28]. For the GTH pseudopotentials, the valence states considered were $1s^2\,2s^1$ for Li, $3s^2\,3p^5$ for Cl, and $4s^2\,4p^5$ for Br [29]. Initial convergence runs were performed to determine the optimal computational parameters. A plane-wave energy cutoff of 796 eV was established, and a Monkhorst–Pack [30] k-point mesh of 8×8×8 was employed. Consequently, the structural geometry optimization of cubic LiSnX₃ (X = Br and Cl) was carried out using these parameters.

The calculations of optical properties necessitate a denser k-mesh; thus, a 14×14×14 mesh was employed. Geometry optimizations were conducted under various pressure conditions, and the band structures of LiSnX$_3$ (where X = Br and Cl) were computed along distinct symmetry lines: Γ (0.0, 0.0, 0.0) − X (0.5, 0.0, 0.0) − M (0.5, 0.5, 0.0) − R (0.5, 0.5, 0.5) at multiple pressure values. The fundamental energy band gap, as well as the gaps at other high-symmetry points, were calculated. Band structures were initially computed at ambient pressure (P = 0) and subsequently at increasing pressure increments of 1 GPa. It was observed that the fundamental energy band gap approaches zero at pressures of 3.5 GPa for LiSnBr$_3$ and 5.7 GPa for LiSnCl$_3$, thus limiting the upper pressure range for the study to these values.

Moreover, (TDOS and PDOS) were calculated for each pressure increment. The investigation of optical properties required the calculation of the frequency-dependent linear optical dielectric function. The real and imaginary components of the dielectric function spectra were computed, and other optical parameters were derived using (DFPT) [31] and Kramers-Kronig relations [32]. The procedure began with the computation of the optical conductivity spectrum as the first step. The frequency-dependent optical conductivity $\sigma(\omega)$ is:

$$\sigma(\omega) = \sigma_1(\omega) + i\sigma_2(\omega) \tag{1}$$

The Kubo-Greenwood (KG) formula can be used to obtain the real part $\sigma_1(\omega)$ of the optical conductivity [33]

$$\sigma_1(\omega) = \frac{2\pi}{\Omega}\sum_{ij} F_{ij}|D_{ij}|^2 \delta\left(\epsilon_i - \epsilon_j - \omega\right) \tag{2}$$

where $\Omega$ is the volume of the unit cell.

$$F_{ij} = [F(\epsilon_i) - F(\epsilon_j)]/\omega \qquad (3)$$

with $F$ being Fermi-Dirac distribution and $D_{ij}$ are the matrix elements of velocity dipole. The $\in_i$ and $\psi_i$ are the energy and wave functions of the $i^{th}$ Kohn-Sham orbital.

$$D_{ij} = \frac{1}{3} \sum_\alpha |\langle \psi_i \nabla_\alpha \psi_j \rangle|^2 \qquad (4)$$

The imaginary part $\sigma_2(\omega)$ part of the optical conductivity follows from the Kramers-Kronig relation:

$$\sigma_2(\omega) = -\frac{2}{\pi} P \int \frac{\sigma_1(v)\omega}{(v^2 - \omega^2)} dv \qquad (5)$$

where $P$ is the principal value of the integral. The real and imaginary components of the frequency-dependent dielectric function are calculated respectively:

$$\epsilon_1(\omega) = 1 - \frac{4\pi}{\omega} \sigma_2(\omega) \qquad (6)$$

$$\epsilon_2(\omega) = \frac{4\pi}{\omega} \sigma_1(\omega) \qquad (7)$$

The dielectric function:

$$\epsilon(\omega) = \epsilon_1(\omega) + i\epsilon_2(\omega) \qquad (8)$$

can also be given by:

$$\epsilon(\omega) = [n(\omega) + ik(\omega)]^2 \qquad (9)$$

where index $n(\omega)$ and $k(\omega)$ are frequency-dependent refractive index and extinction coefficient.

From the above equations $n(\omega)$ can be obtained:

$$n(\omega) = \frac{1}{\sqrt{2}} \sqrt{|\epsilon(\omega)| + \epsilon_1(\omega)} \qquad (10)$$

Then, the absorption coefficients $\alpha(\omega)$ is computed from:

$$\alpha(\omega) = \frac{4\pi}{n(\omega)} \sigma_1(\omega) \qquad (11)$$

For the incident energy photons from 0.008 eV to 41 eV, the real and imaginary parts of dielectric function, refractive index, absorption coefficient and wavelength of $LiSnX_3$ (where X = Br and Cl) under different pressures have been computed.

## 3. Result and discussion

### 3.1. Structural properties

Geometry optimization was conducted to calculate the total energy of $LiSnX_3$ (X = Br and Cl) across different lattice constants, thereby examining the variation of the unit cell volume. The corresponding plots are described in Fig 1. The investigation of the data revealed the unit cell volume $V_o$, and consequently, the lattice constant $a_o \equiv a$, at which the total energy $E_o$ is

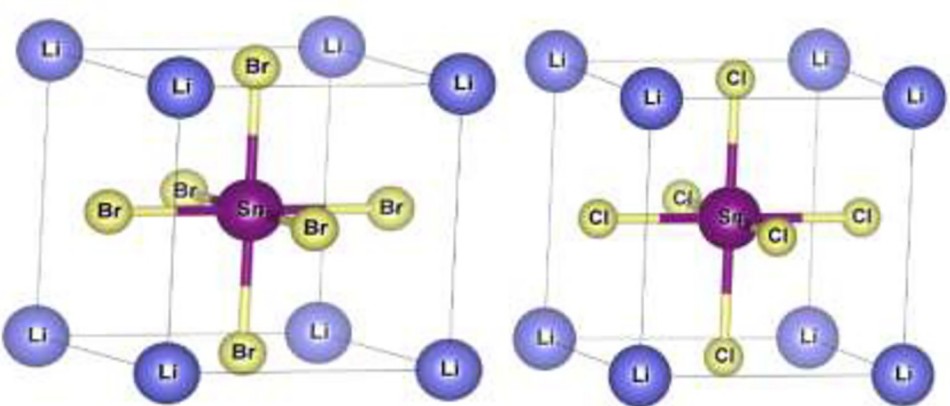

**Fig 1. Ideal cubic structure of LiSnX$_3$ (X = Br, and Cl).**

minimized. Table 1 presents the computed values of $a$ and $E_o$ of LiSnX$_3$ (X = Br and Cl), which are in a good agreement with the available theoretical results. However, no experimental data are available for comparison. The relationship between the total energy E and the volume of the unit cell V was modeled using the third-order Birch–Murnaghan equation of state [34]:

$$E(V) = E_0 + \frac{9V_0 B_0}{16} \left\{ \left[ \left( \frac{V_0}{V} \right)^{\frac{2}{3}} - 1 \right]^3 B_0' + \left[ \left( \frac{V_0}{V} \right)^{\frac{2}{3}} - 1 \right]^2 \left[ 6 - 4 \left( \frac{V_0}{V} \right)^{\frac{2}{3}} \right] \right\} \tag{12}$$

Where $B_o$ and $B_o'$ are the bulk modulus and its pressure derivative, respectively, $V_o$ is unit cell volume, and $E_o$ is the total energy of the unit cells, all at zero pressure. The values of $B_o$ and $B_o'$ for the two structures were obtained from the fittings, shown in Table 1, and are in a reasonable agreement with other results.

Interestingly, the bulk modulus of LiSnBr$_3$ is smaller than that of LiSnCl$_3$. Furthermore, there is an inverse relationship between the atomic number of the halogen atoms and the bulk modulus: as the atomic number increases, the bulk modulus tends to decrease. This observed trend may be attributed to the influence of the halogen atoms on the bonding between Li, Sn, and the halogen atoms in the crystal lattice structures, affecting the overall compressibility and stability of the perovskite's crystal structures.

Fig 2, illustrates how the unit cell volume decreases with increasing pressure for the two studied perovskites. It is evident from the graph that as pressure rises, the unit cell volume declines. Throughout this study, different pressure levels were applied to LiSnX$_3$ (where X = Br and Cl) until their respective band gaps reached zero. Consequently, each perovskite structure exhibited a specific pressure threshold at which this phenomenon occurred. The

**Table 1. Lattice constant (a), bulk modulus (B), pressure derivative of bulk modulus (B') and total energy (Eo) of cubic LiSnX3 (X = Br, and Cl) at P = 0.**

| Compounds | Arrangement | $a$ (Å) | $B$ (GPa) | $B'$ (GPa) | $E_o$(eV) |
|---|---|---|---|---|---|
| LiSnBr$_3$ | Present Work | 5.82 | 19.15 | 4.32 | -102.32 |
| | Experimental | n.a. | n.a. | n.a. | n.a. |
| | Theoretical | 5.72 [35] | 26.15 | 4.05 | n.a. |
| LiSnCl$_3$ | Present Work | 5.58 | 22.27 | 4.29 | -111.69 |
| | Experimental | n.a. | n.a. | n.a. | n.a. |
| | Theoretical | 5.60 [35] | 22.48 | 4.14 | n.a. |

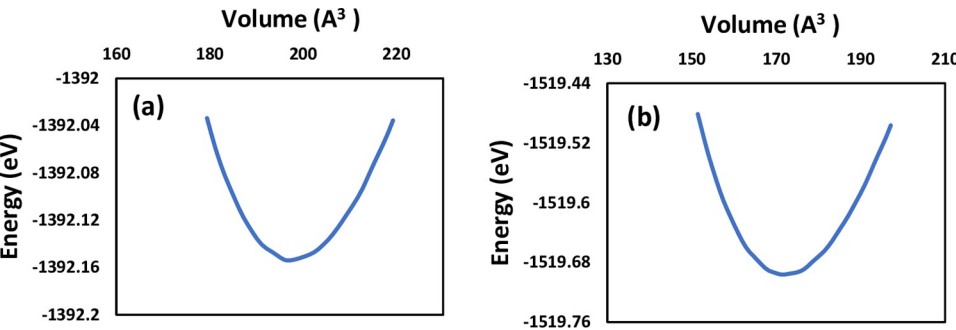

**Fig 2.** Total energy versus the unit cell volume for (a) LiSnBr$_3$ and (b) LiSnCl3 at zero pressure.

lattice constants derived from the unit cell volume, as shown in Fig 3, served as the basis for all subsequent investigations of optoelectronic properties at the corresponding pressure points.

Total volume deformation potential ($a_v^{total}$) due to applying hydrostatic pressure is [36]:

$$a_v^{total} = \frac{dE_{tot}}{dlnV} \tag{13}$$

where $dlnV = \Delta V/V$ with $\Delta V$ is the difference between unit cell volumes at $P_1$ and at $P_2$ and $V$ is the average of the two volumes [37]. For the two perovskites, the total volume deformation potential ($a_v^{total}$) due to applying pressure from 0 GPa to 2 GPa and from 0 to the metallization pressure (P $_{metallization}$) was investigated and gathered in Table 2. It is clear from the table that under similar or the same pressure (0 to 2 GPa), the yield of LiSnBr$_3$ is harder (indicating less deformation potential) than that of LiSnCl$_3$. The table also reveals that a higher pressure is required for the metallization of LiSnCl$_3$, while less pressure is needed for LiSnBr$_3$ to become metallic. These different responses to pressure depend on how tightly the outermost halogen p-orbital (Cl-3p and Br-4p) is bound to its atom. This idea is discussed in more detail in Section 3.3.

The energy formation versus different pressures for LiSnX$_3$ (X = Br and Cl) are plotted in Fig 4. The formation energy represents the energy required to form the perovskite structure from its constituent elements in their respective standard states. A higher (less negative) formation energy implies that the structure is less stable and more likely to decompose into its constituent elements. In contrast, a lower (more negative) formation energy indicates a more thermodynamically stable structure, as it suggests that the compound releases energy upon

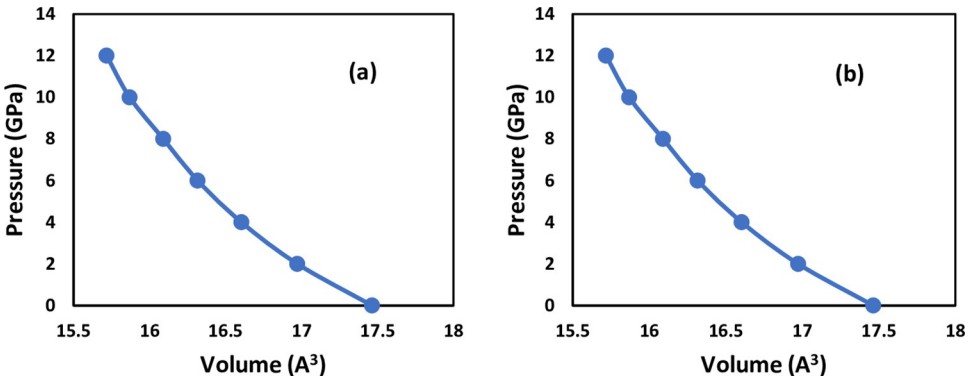

**Fig 3.** Unit cell volume versus hydrostatic pressure for (a) LiSnBr$_3$ and (b) LiSnCl$_3$.

**Table 2. Calculated total volume deformation potential ($a_{vtotal}$).** $P_{metaliz}$ is the metallization pressure.

| Compound | $P_{\text{metalliz}}$ (GPa) | $a_v^{total}$ (eV) | |
|---|---|---|---|
| | | $0 \rightarrow 2$ GPa | $0 \rightarrow P_{\text{metalliz}}$ |
| $LiSnBr_3$ | 4 | -1.35 | -2.07 |
| $LiSnCl_3$ | 6 | -1.19 | -2.54 |

formation, favoring its existence. In addition, the formation energy can provide insights into the response of materials to external conditions such as temperature and pressure. For example, perovskites with low formation energy are more likely to retain their structure and properties under varying conditions. Under ambient conditions, the formation energy can be affected by factors such as ionic size, electronegativity differences between the constituent elements, and lattice distortions. When pressure is applied to $LiSnX_3$ (X = Br and Cl), they undergo structural variations that can significantly change their formation energy. For $LiSnX_3$, applying pressure typically leads to a reduction in lattice volume, which can either stabilize or destabilize the perovskite structure depending on the nature of the X-site halide (Br or Cl). According to the obtained data, $LiSnBr_3$ is more stable than $LiSnCl_3$. The larger ionic size and higher polarizability of Br contribute to a less strained, more stable lattice structure, resulting in more favorable thermodynamic properties. Br is less electronegative and more polarizable than Cl. This can lead to different electronic environments in the crystal structure. The increased polarizability of Br leads to a more stable electronic structure in the perovskite, contributing to a lower formation energy. It is clear from the figures that the stability of the studied perovskites decreases as pressure increases because the formation energy typically increases due to factors such as crystal structure, bond length, changes in electronic structure, and lattice strain. These factors increase the formation energy, leading to increased instability, higher internal energy, changes in bonding characteristics, and increased strain energy. The formation energy of $LiSnX_3$ (X = Br and Cl) are computed using below formula:

$$\Delta E_f = \frac{1}{N} \left[ E_{t(LiSnX_3)} - (E_{Li} + E_{Sn} + 3Ex) \right] \tag{14}$$

Where N is the number of atoms in the unit cell, $E_f$ is the energy formation, Et is the total energy of $LiSnX_3$ and ($E_{Li}$, $E_{Sn}$, $E_{Br}$ and $E_{Cl}$) are the energies of individual atoms in the unit cell.

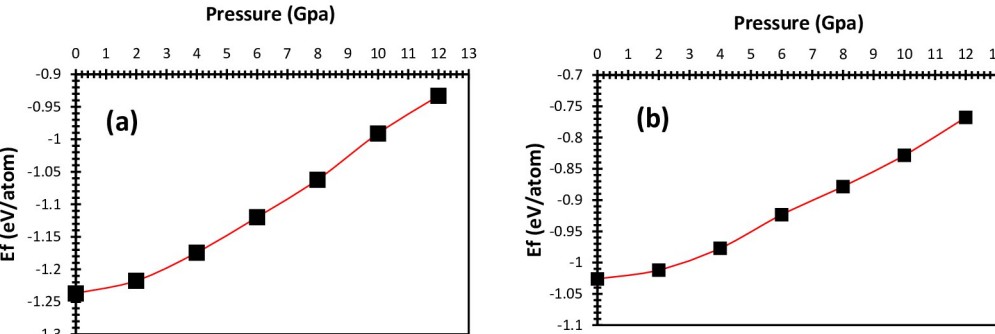

**Fig 4.** Formation energy against hydrostatic pressure of (a) $LiSnBr_3$ and (b) $LiSnCl_3$.

## 3.2 Electronics properties

Investigating the band structure and density of states of perovskites is essential for understanding the electronic properties of these compounds. Substituting any atom in the $ABX_3$ structure, such as in $LiSnX_3$ (where X = Br or Cl), leads to variations in the band structure due to differences in atomic number and electron configuration among the halogen atoms. These variations result in unique properties. Furthermore, applying hydrostatic pressure changes the band structure and, consequently, the properties of the perovskites. Studying the band structures of $LiSnX_3$ (with X being Br or Cl) under various pressures allows us to determine how their properties change with pressure. This insight facilitates the manipulation of properties for different technological applications. The calculated band structures of these compounds at different pressures, as presented in Fig 5, demonstrate that, regardless of pressure, the compounds exhibit a direct fundamental band gap located at the symmetry point R. The calculated GGA-PBE fundamental band gap $E^{R-R}$ for $LiSnBr_3$, and $LiSnCl_3$, at ambient pressure P = 0, are 0.48 eV, 0.92 eV, respectively, see Table 3. The band gaps $E^{M-M}$, $E^{X-X}$ and $E^{\Gamma-\Gamma}$ at the

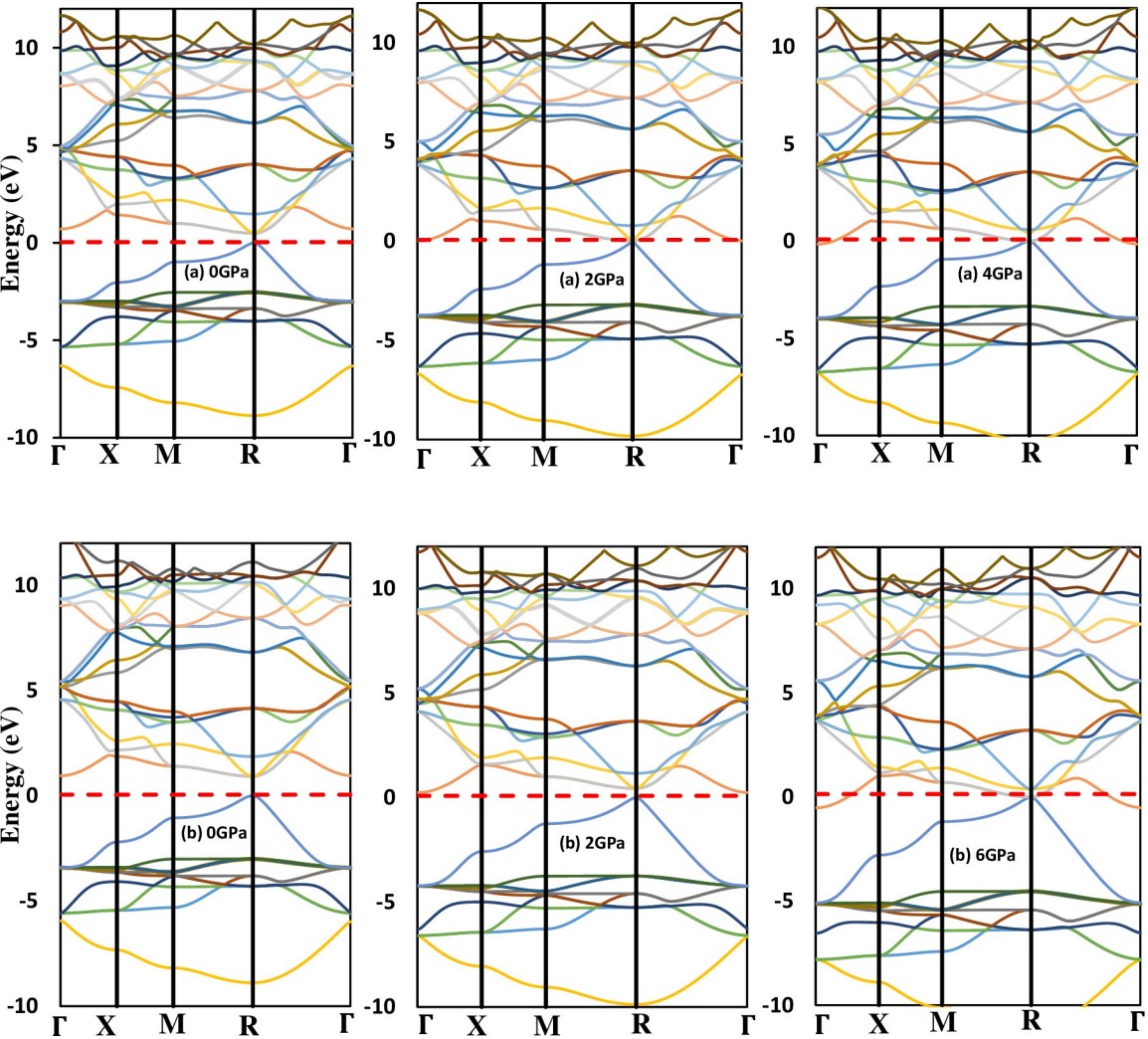

**Fig 5.** Band structure of cubic (a) $LiSnBr_3$ and (b) $LiSnCl_3$ under various pressures. The red dashed line indicates the location of the Fermi level.

**Table 3. The computed energy band gap of LiSnX3 (X = Br and Cl) at various pressures.**

| $E^{gap}$ | LiSnBr$_3$ | | LiSnCl$_3$ | |
|---|---|---|---|---|
| | This work | Other Work | This work | Other Work |
| $E^{R-R}(0GPa)$ | 0.48 | n. a | 0.92 | n. a |
| $E^{R-R}(2GPa)$ | 0.0061 | n.a. | 0.44 | n.a. |
| $E^{R-R}(4GPa)$ | 0 | n.a. | 0.04 | n.a. |
| $E^{R-R}(6GPa)$ | —— | n.a. | 0 | n.a. |

symmetry points M, X and Γ, respectively, are also calculated and shown in Table 3. There are neither practical results nor theoretical investigations of the band gaps for comparison.

Fig 5, show that, at a given pressure, when the halogen atom Cl is replaced by Br in LiSnX$_3$, the band structure changes, which is clearly due to the difference in the atomic number and hence the electron configuration of the halogen atoms. The figures also reveal the change in the band structure of each of the two compounds with the applied pressure. By changing the halogen atom and/or applying pressure, one can control the band structure and (DOS). For example, by controlling the pressure and the halogen atom, it is possible to tune the value of band gap and achieve specific properties to meet the requirements of technological application. These findings have significant implications for the design of optoelectronic devices such as solar cells and light-emitting diodes, where the band gap value is a crucial parameter in determining the performance of devices.

It is noteworthy that the ranges of pressure applied to the two compounds LiSnBr$_3$ and LiSnCl$_3$ are different. The calculations showed that the pressure at which the band gap value becomes zero for each compound is different. The fundamental band gap of LiSnBr$_3$ becomes zero under 4 GPa, while it is 6 GPa for LiSnCl$_3$. Thus, LiSnBr$_3$ was exposed to pressures from 0 GPa to 4 GPa, while LiSnCl$_3$ was subjected to pressures ranging from ambient pressure to 6 GPa. In other words, the study showed that cubic LiSnBr$_3$ and LiSnCl$_3$ are converted from semiconductors to conductors at the pressures of 4 GPa and 6 GPa, respectively. This is an important result as it highlights the significance of the applied pressures as an effective parameter for controlling the properties of the perovskites.

## 3.3 Density of states

The focus of the current study lies in investigating LiSnX$_3$ under different pressures to understand the effect of compression on the electronic properties of the investigated perovskites. The use of the (DOS) to investigate the electronic properties of perovskites is a well-established concept. Specifically, this study studies the electronic properties under various pressures, a task for which knowledge of the (TDOS) and the (PDOS) is essential. The TDOS and PDOS for cubic LiSnX$_3$ (X = Br and Cl) at three different pressures were calculated and are presented in Fig 6. These plots provide a comprehensive view of the changes in the number of electron states as pressure is increased, offering valuable insights into the behavior of the material under compression.

For the two cubic LiSnX$_3$ perovskites (where X = Br and Cl), the projected density of states (PDOS) shown in Fig 7 indicates that the lower region of the valence band (VB) ($< -10$ eV) arises primarily from substantial contributions of Li-4p, Sn-$4s$, and halogen atom-s orbitals, along with a minor contribution from Sn-3d orbitals. Conversely, the higher region of the VB ($> -10$ eV) is dominated by significant contributions from Sn-4p, Sn-4s, and halogen atom-p orbitals, with the halogen atom-p orbital contribution being particularly pronounced. The contributions of Li-3d and Sn-3d orbitals are minimal. This tendency remains consistent

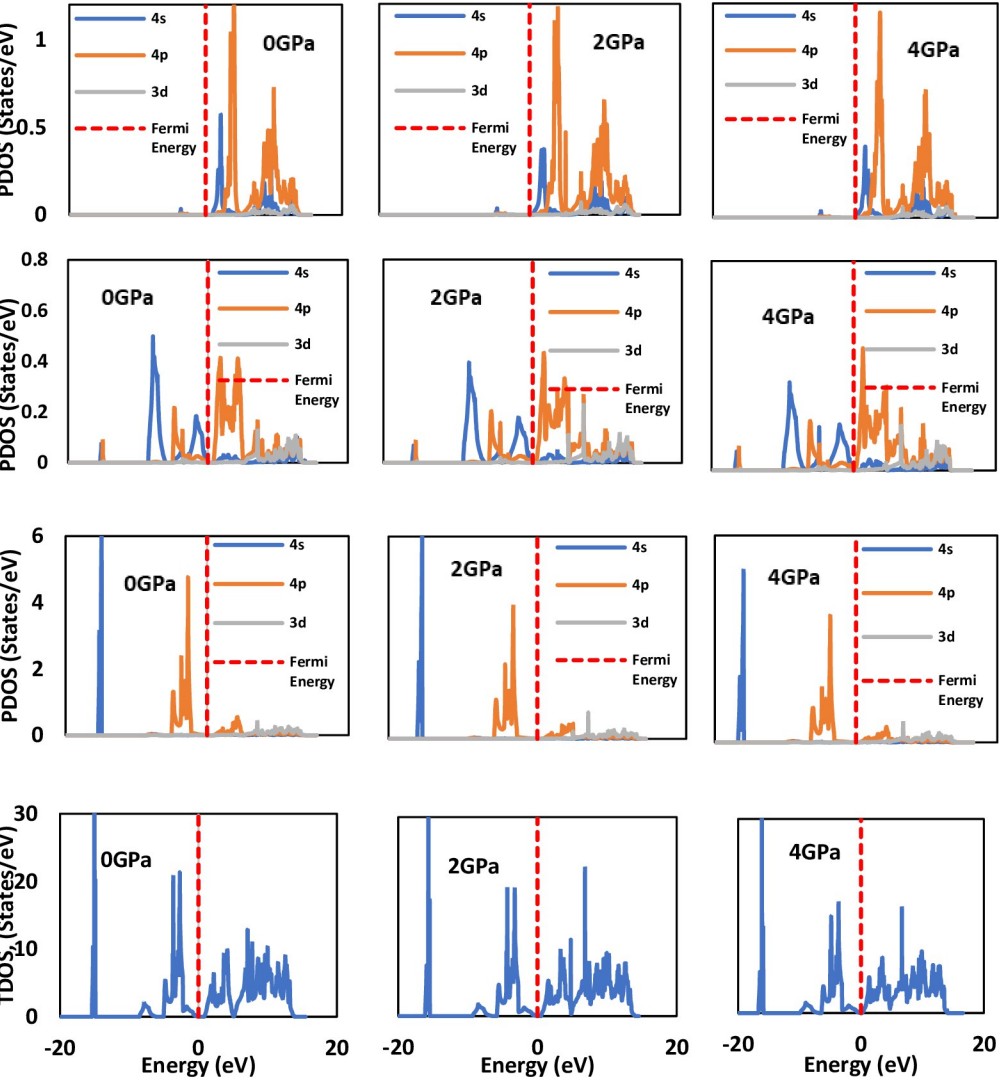

**Fig 6. Total and partial densities of states for LiSnBr₃ under three different pressures.** The red dashed line
indicates the location of the Fermi level.

across all pressures. The s, p, and d orbitals of both Li and Sn are the primary contributors to
the conduction band (CB), with Li-3d playing a more prominent role. In contrast, the contri-
butions from the p and d orbitals of the halogen atom are considerably less significant, and the
contribution from the s orbital is almost negligible. The higher region of the CB is predomi-
nantly formed by the Li-4d orbital, while the lower region is primarily composed of the Sn-4p
orbital. The PDOS of the halogen atoms (Br and Cl) are closely aligned, indicating that substi-
tuting one for the other has minimal impact. However, the TDOS of the high VB (approxi-
mately –10 eV) is greater at 0 GPa. The results clearly demonstrate that applying pressure
pushes the VB closer to the Fermi level, thereby facilitating electron transitions to the CB and
enhancing the conductivity of the perovskites.

The calculated PDOS reveals that the edge of the valence band primarily arises from the 3p
orbital of Cl in LiSnCl₃ and the 4p orbital of Br in LiSnBr₃. In both compounds, the edge of the
conduction band predominantly originates from the 4p orbital of Sn, as illustrated in Fig 7.

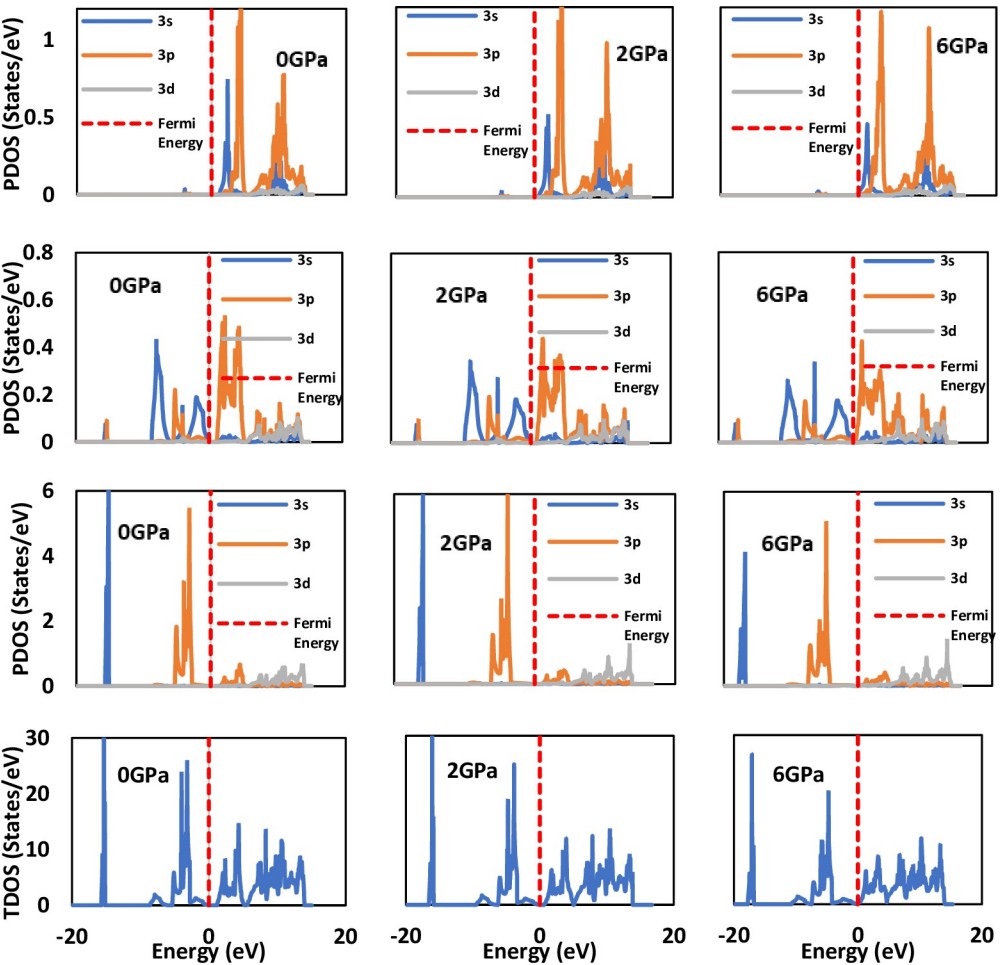

**Fig 7. Total and partial densities of states for LiSnCl₃ under three different pressures.** The red dashed line indicates the location of the Fermi level.

This implies that the energy gap between the conduction and valence bands (R-R energy gap) results from the difference between the 4p orbital of the Sn atom and the p orbitals of the halogen atoms (Br-4p and Cl-3p). Consequently, the first transition from the valence band to the conduction band occurs from the halogen p orbital to the Sn-4p orbital.

Fig 7, also shows that as pressure increases, the 4p orbital of Sn shifts downward, while the p orbitals of the halogen atoms (Br-4p and Cl-3p) shift upward, thereby reducing the R-R gap until they overlap at a certain pressure, known as the metallization pressure. In other words, metallization is caused by the pressure effects on the Sn-4p and halogen p atomic orbitals. The impact of pressure on the R-R energy gap of the two compounds depends on the binding strength of the uppermost halogen p orbital (Br-4p and Cl-3p) to its respective atom. The 3p orbital of Cl is more tightly bound (has lower orbital energy) than the 4p orbital of Br, making it more resistant to applied pressure. Consequently, LiSnCl₃ must metallize at a higher pressure compared to LiSnBr₃. This finding is confirmed by the present results, which indicate that the obtained metallization pressures for LiSnBr₃ and LiSnCl₃ are 4 GPa and 6 GPa, respectively. To illustrate the effect of pressure, the TDOS of LiSnX₃ (where X = Br and Cl) at various pressure values are plotted together in Fig 7. The figure demonstrates that pressure has a substantial effect on the TDOS. As pressure increases, the TDOS curve shifts toward higher

energies, with the top of the valence band shifting upward more significantly than that of the conduction band. Thus, the fundamental energy band gap decreases with increasing pressure, ultimately reaching zero at the metallization point of the perovskites.

## 3.4 Optical properties

Under different pressures, the imaginary part $\varepsilon_2(\omega)$ of the dielectric function of cubic LiSnX$_3$ (X = Br and Cl) as a function of photon energy has been calculated and is presented in Fig 8. Fig 8A and 8B display the imaginary part of the dielectric function for three different pressures. The results indicate that the effect of pressure on the $\varepsilon_2(\omega)$ spectra of the two compounds corresponds with the behavior observed in the fundamental energy band gap; specifically, $\varepsilon_2 = 0$ for photon energies less than the gap. For instance, at zero pressure, the spectra show values at approximately 0.1 eV for LiSnBr$_3$ and 0.35 eV for LiSnCl$_3$, as noted in Table 3. As pressure increases, the $\varepsilon_2$ spectrum of the two compounds exhibits a shift to lower photon energies, reflecting the decrease in the band gap with the applied pressure. The spectra of the imaginary part of the dielectric function, $\varepsilon_2$, for LiSnCl3 exhibit two peaks, as shown in Fig 8B. The first peak (1) occurs close to 1 eV. Consulting the DOS in Fig 6, it is evident that this peak corresponds to the energy levels of the 4s and 4p orbitals of Sn. Thus, peak (1) arises from electron transitions from the 4s orbital of Sn to the 4p orbital of Sn. The second peak (2) results from electron transitions from the 3p orbital of Cl to the 3d orbital of Li. This peak is located at 9 eV, which aligns with the energy levels of the 3p-Cl and 3d-Li orbitals, as depicted in Fig 6. A similar pattern is observed for LiSnBr$_3$.

An important observation is the inverse relationship between the band gap and the imaginary part of the dielectric function, $\varepsilon_2$, for the two perovskites. Remarkably, an increase in applied pressure leads to a decrease in the band gap, which subsequently results in an increase in the imaginary part of the dielectric function, as illustrated in Fig 8. The results indicate fluctuations in the imaginary part of the dielectric function for all pressures in LiSnCl$_3$, starting from around 1.5 eV; however, this fluctuation is not observed in LiSnBr$_3$. The imaginary part of the dielectric function is closely related to the band structure and provides a detailed explanation of the absorbance properties of materials. The maximum peaks of the imaginary part of the dielectric function vary with pressure: as pressure increases, these peaks increase, especially in the infrared and visible regions. Beyond the visible region, the imaginary part of the dielectric function decreases. At high photon energies ($>$ 4 eV), the imaginary part of the dielectric function for LiSnX$_3$ (where X = Br or Cl) approaches zero under all applied pressures.

Fig 9, presents the spectra of the real part of the dielectric function $\epsilon_1(\omega)$ plotted against photon energy for LiSnX$_3$ (X = Br and Cl) at different pressures. At ambient pressure, the

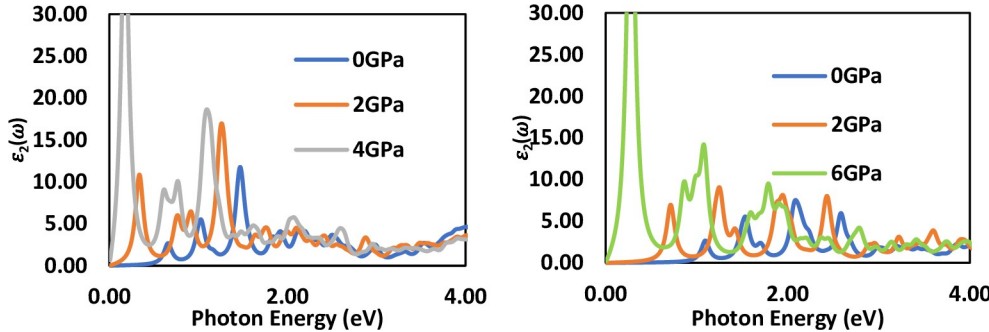

**Fig 8.** Imaginary part of dielectric function spectra of (a) LiSnBr$_3$ and (b) LiSnCl$_3$ under different applied pressures.

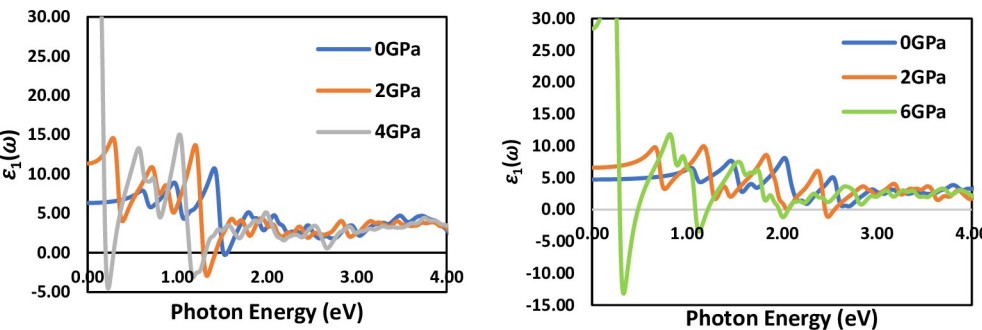

**Fig 9. Real part of dielectric function spectra of cubic LiSnX$_3$ (X = Br and Cl) under different applied pressures.**

zero-frequency limit increases as the halogen changes from Cl to Br. The maximum $\epsilon_1$ (ω) values occur at approximately 1.41 eV and 2 eV for LiSnBr$_3$ and LiSnCl$_3$, respectively. The values of $\epsilon_1$(ω) reach zero at photon energies of 1.66 eV and 2.8 eV for LiSnBr$_3$ and LiSnCl$_3$, respectively. Beyond these energies, $\epsilon1$(ω) becomes negative, resulting in higher reflectivity of the materials. After 2.5 eV, the real part of the dielectric function for both compounds decrease and approaches unity, causing them to become transparent to high-energy radiation and invisible to the naked eye [38].

The static dielectric constant $\epsilon_1$(0) (see Table 4) and its relationship with pressure and energy band gap in LiSnX$_3$ (X = Br and Cl) represent a novel area of research that provides insights into the electronic and optical properties of these materials. The static dielectric constant increases with pressure, exhibiting an inverse relationship with the energy band gap—a characteristic that is unique to these perovskites and has not been extensively studied in other materials. A prominent observation in this study is the shift of the maximum peaks of $\epsilon_1$(0) toward lower photon energies as pressure increases. This shift indicates a change in the electronic and optical properties of these materials with pressure, which is an important finding for the development of pressure-dependent optoelectronic devices. Furthermore, the observation that the energy band gap of LiSnBr$_3$ is smaller than that of LiSnCl$_3$, along with the corresponding real part of the dielectric constant reaching zero at lower energies for LiSnBr$_3$ compared to LiSnCl$_3$, highlights a unique property of this material that has not been reported previously [39, 40].

The behavior of the refractive index spectrum is similar to that of the real part of the dielectric constant. In Fig 10, the energy of projected photons is plotted against the refractive index n(ω) of LiSnX$_3$ (X = Br and Cl). It is observed that the refractive index of these compounds increases from Cl to Br, as shown in Table 4. Furthermore, the figure illustrates that the refractive index increases with increasing pressure, indicating that this enhancement makes the perovskites highly suitable for photonic applications. At 0 GPa, the zero-frequency refractive index (n (0)) of LiSnBr$_3$ and LiSnCl$_3$ are 2.52 and 2.17, respectively. However, the refractive index at zero frequency is not significantly different from that at very low frequencies, as there is no abrupt change when the frequency approaches zero. The refractive index represents the

**Table 4. Computed static dielectric constant and refractive index, along with maximum real part of dielectric function and maximum refractive index, all at ambient pressure.** The incident photon frequency $v$ corresponding to each maximum value is also given.

| Compound | $\varepsilon_1(0)$ | $n$ (0) | $\varepsilon_{1max}$ (ω) | $v$ (×10$^{14}$ Hz) | $n_{max}$ (ω) | $v$ (×10$^{14}$ Hz) |
|---|---|---|---|---|---|---|
| LiSnBr$_3$ | 6.32 | 2.51 | 10.73 | 3.42 | 3.43 | 3.46 |
| LiSnCl$_3$ | 4.71 | 2.17 | 8.11 | 4.9 | 2.96 | 4.94 |

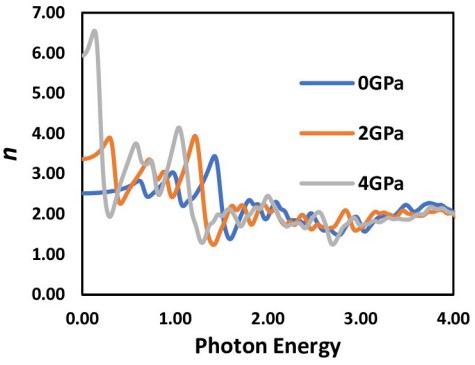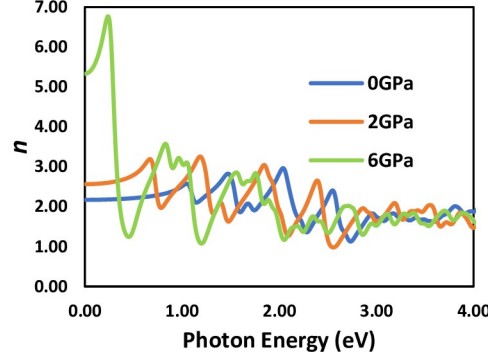

**Fig 10. Refractive indices of LiSnBr₃ and LiSnCl₃.**

factor by which a wave slows down relative to the speed of light in a vacuum ($c$). This enables the calculation of the velocity ($v$) of an electromagnetic wave as follows [39, 41]:

$$v = \frac{c}{n}$$

The response of materials to incoming electromagnetic radiation is described by the frequency-dependent complex refractive index: $n^*(\omega) = n(\omega) + i\kappa(\omega)$. The imaginary part, $\kappa$ (called the extinction coefficient), measures how much radiation of a given frequency is absorbed by the medium, while the real part, $n$ (called the refractive index), is related to the phase velocity of the radiation in the medium [42]. Essentially, $n = c/v$ where $v$ is the phase speed of the radiation in the medium and c is the speed of light in a vacuum. The only way for the value of $n$ to be less than one is when the phase velocity $v$ in the medium exceeds the speed of light $c$. This does not violate the theory of special relativity because the phase velocity is the velocity of a single-frequency wave, which does not carry energy or information. In other words, a phase velocity exceeding $c$ does not imply the propagation of signals at a speed greater than $c$. Fig 10, shows that the refractive indices of LiSnBr₃ and LiSnCl₃ increase as the incident photon energy increases, reaching a maximum at 3.43 eV and 2.96 eV, respectively, under ambient pressure (see Table 4). The refractive indices then gradually decrease to 1.5 around 1.46 eV and 2.86 eV, respectively, and fall below unity for higher incident photon energies. Thus, the present results indicate that incoming radiation with energies greater than 1.46 eV and 2.86 eV propagates with $v > c$ through LiSnBr₃ and LiSnCl₃, respectively.

The absorption coefficient is a crucial parameter for understanding how materials absorb light energy. It provides valuable information about the ability of materials to convert solar energy, which is directly applicable in fields such as solar cells and electromagnetic device development. The absorption spectrum of LiSnX₃ (X = Br and Cl) as a function of photon energy under different pressures is plotted and shown in Fig 11. The figure indicates that the absorption edge begins at approximately 0.48 eV and 0.64 eV for LiSnBr₃ and LiSnCl₃, respectively. The maximum absorption coefficient is observed at around 12.92 eV and 13.74 eV under pressures of 4 GPa and 6 GPa for LiSnBr₃ and LiSnCl₃, respectively. The absorption coefficients obtained for the studied perovskites are consistent with the band gap values calculated from the electronic band structure, thereby confirming the reliability of the present work. A similar trend is observed in the imaginary part of the dielectric constant. The high peaks in the absorption spectrum indicate the regions of maximum light energy absorption. It is evident from the figure that the peak positions shift from Br to Cl, suggesting that both the

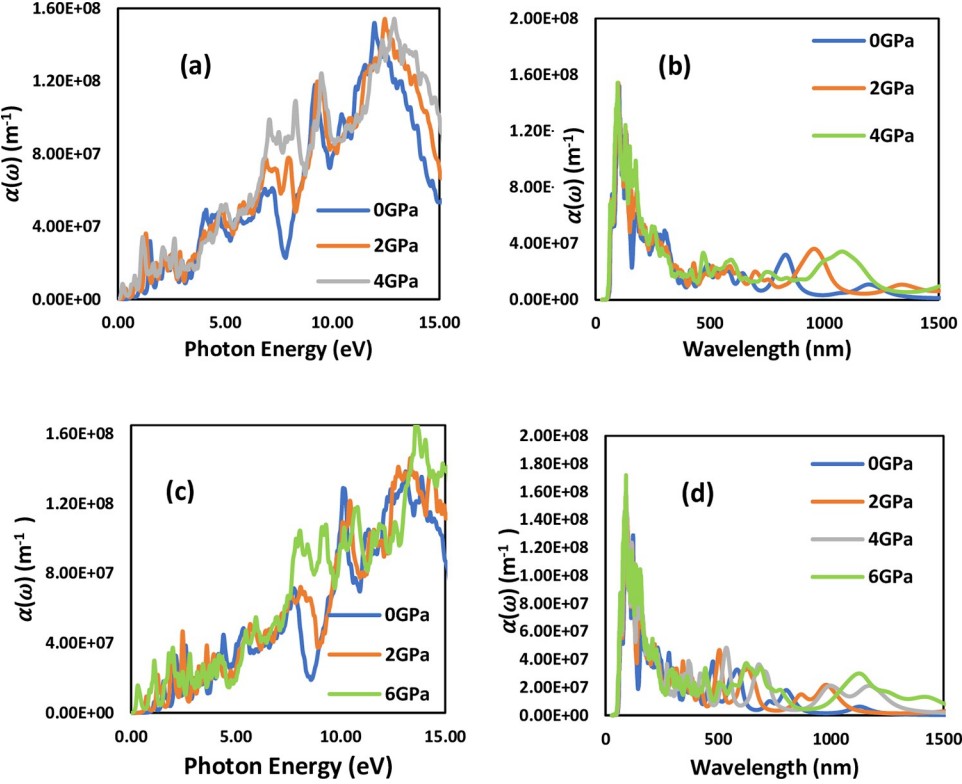

**Fig 11.** Absorption coefficient spectrum versus photon energy (a) LiSnBr$_3$, (b) LiSnCl$_3$ and wavelength (c) LiSnBr$_3$, (d) LiSnCl$_3$, with enhance pressures.

halogen atoms and the applied pressure significantly influence the peak positions. The highest absorption peaks are associated with higher pressures.

Fig 11, demonstrates that perovskites show a high absorption capability at minimum energy levels, making the studied compounds highly suitable for solar energy applications. In additions, it is observed that increasing pressure on these perovskites enhances their absorbance significantly, without causing substantial changes in their energy levels. These results provide clear evidence that pressure is a crucial factor in improving the absorption capacity of these materials. Consequently, pressure should be considered an effective parameter for enhancing the performance of perovskites, making them valuable for various applications. To further understand the optical properties of the studied perovskites, the wavelength versus absorption coefficient for LiSnX$_3$ is plotted in Fig 11. The plot shows that LiSnCl$_3$ exhibits a maximum absorption peak in the visible range, around 600 to 700 nm. However, both perovskites show a maximum absorption peak in the approximate range of 100 to 200 nm, which lies in the ultraviolet (UV) region of the spectrum. Since the absorption peak of LiSnBr$_3$ falls within the UV range, it can potentially be used in medical applications, such as the decontamination of surgical equipment. Fig 11 also indicates that the absorption of LiSnX$_3$ (X = Br and Cl) increases with increasing pressure.

The investigation of the structural and optoelectronic properties of cubic lead-free halide perovskites, specifically LiSnX$_3$ (X = Br and Cl), under hydrostatic pressure presents a compelling research area due to the potential of these materials for next-generation optoelectronic applications. Lead-free halide perovskites are particularly attractive owing to their reduced toxicity and environmental impact compared to lead-containing counterparts. By applying

hydrostatic pressure, the electronic band structure can be systematically tuned to study phase transitions, providing insights into their stability and electronic behavior under varying pressures. Understanding how these properties evolve with pressure not only deepens our fundamental knowledge of perovskite materials but also aids in designing more efficient and robust optoelectronic devices, such as solar cells, LEDs, and photodetectors. This research bridges the gap between theoretical predictions and practical applications, paving the way for safer, more sustainable technologies in the rapidly advancing field of materials science.

## 4. Conclusion

In this research, the structural and optoelectronic properties of lead-free halide perovskites, $LiSnX_3$ (X = Br and Cl), were investigated under the influence of hydrostatic pressure using the PBE approach within the GGA exchange-correlation energy framework of DFT. According to the obtained electronic properties, $LiSnX_3$ (X = Br and Cl) show a direct band gap at the high-symmetry k-point R. The calculated formation energies confirm that $LiSnBr_3$ is more stable than $LiSnCl_3$. Additionally, the band gap of $LiSnX_3$ (X = Br and Cl) decreases with increasing pressure. Using DFPT and the Kramers-Kronig relation, the optical properties of $LiSnX_3$ were calculated. The results show that, as pressure increases, the absorption coefficient and refractive index of the studied perovskites also increase, shifting toward higher photon energies. A similar trend is observed when substituting Cl with Br, although the shift occurs toward lower photon energies. At ambient pressure, both perovskites exhibit a metallic response to photon energies in the ranges of 9–14 eV for $LiSnBr_3$ and 10–14 eV for $LiSnCl_3$. As pressure increases, these ranges shift toward higher photon energies. For both perovskites, the zero-frequency refractive index increases with rising pressure and also increases when halogen substitution changes from Cl to Br at a given pressure. The results further indicate that the absorption of $LiSnX_3$ increases with increasing pressure, particularly in the visible spectrum, with the absorption coefficient being approximately four times higher in the low-energy UV region. These findings suggest that $LiSnX_3$ (X = Br and Cl) are promising materials for optoelectronic devices and solar cell technologies. To the best of the author's knowledge, this is the first theoretical study on the structural and optoelectronic properties of $LiSnX_3$ (X = Br and Cl) under hydrostatic pressure, facilitating for future research in this field.

## Acknowledgments

The authors would like to thank the University of Zakho and Physics Department for the assistance for the completion of this work.

## Author Contributions

**Formal analysis:** Mohammed Noor S. Rammoo.

**Software:** Bewar M. Ahmad.

**Writing – original draft:** Hameed T. Abdulla.

**Writing – review & editing:** Nawzad A. Abdulkareem.

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
