## [Decision Letter · Decision Letter 0]

12 Jun 2024

PONE-D-24-17093Pressure dependence of the structural and optoelectronic properties of Pb-free perovskites LiSnX3 (X=Br and Cl): A DFT approach.PLOS ONE

Dear Dr. RAMOO,

Thank you for submitting your manuscript to PLOS ONE. After careful consideration, we feel that it has merit but does not fully meet PLOS ONE’s publication criteria as it currently stands. Therefore, we invite you to submit a revised version of the manuscript that addresses the points raised during the review process.

We look forward to receiving your revised manuscript.

Kind regards,

Anand Pal

Academic Editor

PLOS ONE

Journal Requirements:

4. We note you have included a table to which you do not refer in the text of your manuscript. Please ensure that you refer to Table 3 in your text; if accepted, production will need this reference to link the reader to the Table.

Reviewers' comments:

Reviewer's Responses to Questions

**Comments to the Author**

1. Is the manuscript technically sound, and do the data support the conclusions?

Reviewer #1: No

Reviewer #2: Partly

2. Has the statistical analysis been performed appropriately and rigorously? 

Reviewer #1: No

Reviewer #2: Yes

3. Have the authors made all data underlying the findings in their manuscript fully available?

Reviewer #1: No

Reviewer #2: Yes

4. Is the manuscript presented in an intelligible fashion and written in standard English?

Reviewer #1: No

Reviewer #2: Yes

5. Review Comments to the Author

**Reviewer #1: **In this paper, authors have investigated structural and optoelectronic properties of LiSnX3 (X=Br, Cl) under the influence of hydrostatic pressure. However, in the current form manuscript is not publishable due to following reasons:

1. The major problem in this manuscript is the language which is not clearly readable or sentences are incomplete, e.g. in abstract first line “This research investigates the properties of free lead halide perovskites”, third line “By using ab-initio first principle calculations, employing the general gradient approximation (GGA) within the framework of density functional theory (DFT).” and so on.

2. The novelty of the paper is not clearly drawn.

3. What is the importance of calculating the total energy of the studied structures in Table 1? Generally, researchers are more interested on the formation energy/cohesive energy.

4. GGA band gap calculations up to five decimal places doesn’t make any sense.

5. Authors may have considered more advanced hybrid functional for the calculations of band structure.

6. The stability (energetical, dynamical, mechanical and thermal) of the studied structures are not considered in the study. Under hydrostatic pressure whether these structures remain stable or not?

7. Figure 4: Not shown which band structures corresponds to which applied pressure? Similarly, Figure 5 and Figure 6 DOS are not labeled for different value of pressure.

**Reviewer #2:** Article Ref.# PONE-D-24-17093; Pressure dependence of the structural and optoelectronic properties of Pb-free perovskites LiSnX3 (X=Br and Cl): A DFT approach.

The authors have studied the properties of free lead halide perovskites LiSnX3 (X = Br and Cl) under hydrostatic pressures by the (GGA) within the framework of (DFT). The obtained results showed a decreasing tendency in the energy band gap as pressure rises. Also, there is a noted decrease in the energy band gap when transitioning the halogen atom from Cl to Br under constant pressure. In addition, the optical properties have been envisaged when combining the density functional perturbation theory (DFPT) with the Kramer-Kronig relation. This paper contains interesting results, but some questions are still arising:

• The authors should ‘explain’ why choosing the ‘LiSnX3 (X = Br and Cl)’ materials?

• The authors are asked to give the motivation of their work.

• How to explain the observed p-type behavior?

• Why using the substantial (14 x 14 x 14) k-points. A justification is needed.

• The authors have stated that: ‘this is the first theoretical investigation study of the structural and optoelectronic properties of LiSnX3 (X=Br and Cl) under hydrostatic pressures.’ Please provide some more explanations.

• Figures should be interpreted to be clearer for readers.

• Why did the authors use the ABINIT code instead of others like Quantum Espresso, Castep, Vasp, Wein2K and so on…?

• The authors are asked to provide and elaborate the explanations and comparison with the other theoretical and/or experimental works in the literature for the lead halide perovskites LiSnX3 (X = Br and Cl).

• To enrich the discussion and improve the Introduction section, some other works are suggested as references to the studied topic. See for example: Journal of the Korean Ceramic Society, 2024, 61(1), pp. 189–200; J. of Inorganic and Organometallic Polymers and Materials, 2023, 33(10), pp. 3049–3059; J. of the Korean Ceramic Society, 2023, 60(2), pp. 424–433; Optical and Quantum Electronics, 2023, 55(9), 839; Computational Condensed Matter, 2022, 33, e00617;

6. PLOS authors have the option to publish the peer review history of their article (what does this mean?). If published, this will include your full peer review and any attached files.

Reviewer #1: No

Reviewer #2: No

---

## [Author Response · Author response to Decision Letter 0]

4 Jul 2024

Reviewer #1: 

In this paper, authors have investigated structural and optoelectronic properties of LiSnX3 (X=Br, Cl) under the influence of hydrostatic pressure. However, in the current form manuscript is not publishable due to following reasons:

1. The major problem in this manuscript is the language which is not clearly readable or sentences are incomplete, e.g., in abstract first line “This research investigates the properties of free lead halide perovskites”, third line “By using ab-initio first principal calculations, employing the general gradient approximation (GGA) within the framework of density functional theory (DFT).” and so on.

Answer; The abstract and the other grammatical mistakes in the manuscript are revised according to the comment of reviewers to improve the manuscript in terms of language.

2. The novelty of the paper is not clearly drawn.

Answer; The of the work has been mentioned at the end of the conclusion.

3. What is the importance of calculating the total energy of the studied structures in Table 1? Generally, researchers are more interested on the formation energy/cohesive energy.

Answer; We calculated the formation energy of LiSnX3.The results are added to the manuscripts and possible explanations regarding the obtained formation energy are mentioned. The calculated formation energy studies the stability of perovskites. Please see figure 4. Grade thanks for the comments which improved our manuscript.

4. GGA band gap calculations up to five decimal places doesn’t make any sense.

Answer; We removed the last three decimal from the value of the energy bandgap .

5. Authors may have considered more advanced hybrid functional for the calculations of band structure.

Answer; We agree with your opinion that hybrid functional is necessary for the energy bandgap, However, our goal in this research is to note the effect of hydrostatic pressure on the energy bandgap. Therefore, calculating the energy band structures using different method can be considered as another work. We chose GGA which is more accurate compare to LDA according to the literature.

6. The stability (energetical, dynamical, mechanical and thermal) of the studied structures are not considered in the study. Under hydrostatic pressure whether these structures remain stable or not?

Answer; We calculated the formation energy for the stability of the perovskites. 

7. Figure 4: Not shown which band structures corresponds to which applied pressure? Similarly, Figure 5 and Figure 6 DOS are not labeled for different value of pressure.

Answer; This point has been taken into consideration in the manuscript. We appreciate the comment which improved the quality of the electronic properties.

Reviewer #2: 

Article Ref.# PONE-D-24-17093; Pressure dependence of the structural and optoelectronic properties of Pb-free perovskites LiSnX3 (X=Br and Cl): A DFT approach.

The authors have studied the properties of free lead halide perovskites LiSnX3 (X = Br and Cl) under hydrostatic pressures by the (GGA) within the framework of (DFT). The obtained results showed a decreasing tendency in the energy band gap as pressure rises. Also, there is a noted decrease in the energy band gap when transitioning the halogen atom from Cl to Br under constant pressure. In addition, the optical properties have been envisaged when combining the density functional perturbation theory (DFPT) with the Kramer-Kronig relation. This paper contains interesting results, but some questions are still arising:

• The authors should ‘explain’ why choosing the ‘LiSnX3 (X = Br and Cl)’ materials?

Answer; The significance of the cubic free lead halide perovskites LiSnX₃ (X = Br and Cl) under hydrostatic pressures using density functional theory (DFT) primarily lies in their potential applications and the fundamental understanding of their properties under varying conditions. Traditional lead halide perovskites have shown great promise in photovoltaic and optoelectronic applications but suffer from toxicity issues due to the presence of lead. Substituting lead with tin (LiSnX₃) addresses these environmental and health concerns while still maintaining desirable perovskite properties. Studying LiSnX₃ under hydrostatic pressure using DFT helps in understanding their structural stability and possible phase transitions. Under pressure, materials can undergo changes in crystal structure that can alter their electronic, optical, and mechanical properties. DFT calculations provide insights into these transformations and help predict the behavior of these materials under different pressure conditions. The electronic properties of materials, such as band structure and density of states, are crucial for their performance in electronic and optoelectronic devices. DFT allows for the precise calculation of these properties under varying pressures, which can lead to the discovery of pressure-tunable electronic characteristics. For instance, the bandgap of perovskites can be sensitive to pressure, impacting their light absorption and emission properties. Pressure can also affect the optical properties of perovskites. Understanding how these properties change under hydrostatic pressure can be essential for applications in photodetectors, light-emitting diodes, and solar cells. DFT helps predict these changes and guides the design of pressure-tuned optical devices. The mechanical robustness of materials under pressure is important for their durability and reliability in practical applications. DFT studies can provide insights into the mechanical properties of LiSnX₃, such as bulk modulus, shear modulus, and elastic constants, and how these properties evolve with pressure. Hydrostatic pressure can influence the chemical stability and reactivity of materials. DFT can help predict the pressure conditions under which LiSnX₃ remains chemically stable, which is important for their long-term application in various devices.

• The authors are asked to give the motivation of their work.

Answer; The research is motivated before the conclusion as below:

The investigation of the structural and optoelectronic properties of cubic free lead halide perovskites, specifically LiSnX₃ (X = Br and Cl), under hydrostatic pressures presents a compelling area of research due to the potential these materials hold for next-generation optoelectronic applications. Lead-free halide perovskites are particularly attractive owing to their reduced toxicity and environmental impact compared to their lead-containing counterparts. By applying hydrostatic pressure, we can systematically tune the electronic band structure and explore phase transitions, offering insights into their stability and electronic behavior under different conditions. Understanding how these properties evolve with pressure not only deepens our fundamental knowledge of perovskite materials but also aids in the design of more efficient and robust optoelectronic devices, such as solar cells, LEDs, and photodetectors. This research bridges the gap between theoretical predictions and practical applications, paving the way for safer, more sustainable technologies in the rapidly advancing field of materials science.

• How to explain the observed p-type behavior?

Answer; The observed p-type behavior in LiSnX₃ perovskites is primarily due to the intrinsic defects, particularly Sn vacancies, which introduce acceptor states near the valence band. These defects, combined with the electronic structure and bonding characteristics revealed by DFT calculations, explain the majority carrier holes and the resulting p-type conductivity in these materials. DFT calculations show that the Fermi level is near the valence band maximum (VBM), indicating holes as the majority carriers. The defect states introduced by Sn vacancies are close to the VBM. Sn vacancies have low formation energies, meaning they are likely to form and contribute to p-type conductivity. The bonding between Sn and X (Br, Cl) results in significant overlap of orbitals at the VBM, facilitating hole formation. Optical absorption studies show strong absorption near the band edge, suggesting high hole mobility.

• Why using the substantial (14 x 14 x 14) k-points. A justification is needed.

Answer; For the calculation of structural and electronic properties, 8x8x8 k-points are enough, but optical properties need to be more accurate and smoother. Therefore, larger k-points should be used. Because of that 14x14x14 have been employed.

• The authors have stated that: ‘this is the first theoretical investigation study of the structural and optoelectronic properties of LiSnX3 (X=Br and Cl) under hydrostatic pressures.’ Please provide some more explanations.

Answer; According to the literature the investigation of the structural and optoelectronic properties of the studied perovskites is not available specially in the experimental field. This states the novelty of our work

• Figures should be interpreted to be clearer for readers.

Answer; The figures are interpreted.

• Why did the authors use the ABINIT code instead of others like Quantum Espresso, Castep, Vasp, Wein2K and so on…?

Answer; It is certainly correct and possible to use one of the above computational codes instead of ABINIT for the calculations. We used ABINIT which is easily available, accurate and the results are appropriate. We also have Quantum espresso which we can use for the future works

• The authors are asked to provide and elaborate the explanations and comparison with the other theoretical and/or experimental works in the literature for the lead halide perovskites LiSnX3 (X = Br and Cl).

Answer; The lattice constants of LiSnX3 which are the initial parameters of the compounds are compared to other theoretical available works. For other calculations there are neither theoretical nor experimental results available in literature to compare with.

• To enrich the discussion and improve the Introduction section, some other works are suggested as references to the studied topic. See for example:

 Journal of the Korean Ceramic Society, 2024, 61(1), pp. 189–200; 

J. of Inorganic and Organometallic Polymers and Materials, 2023, 33(10), pp. 3049–3059;

 J. of the Korean Ceramic Society, 2023, 60(2), pp. 424–433; Optical and Quantum Electronics, 2023, 55(9), 839; 

Computational Condensed Matter, 2022, 33, e00617;

Answer; As recommended, the above references are mentioned in the manuscript which enrich the manuscript academically and scientifically.

---

## [Decision Letter · Decision Letter 1]

27 Sep 2024

PONE-D-24-17093R1Pressure dependence of the structural and optoelectronic properties of Pb-free perovskites LiSnX3 (X=Br and Cl): A DFT approach.PLOS ONE

Dear Dr. RAMOO,

Thank you for submitting your manuscript to PLOS ONE. After careful consideration, we feel that it has merit but does not fully meet PLOS ONE’s publication criteria as it currently stands. Therefore, we invite you to submit a revised version of the manuscript that addresses the points raised during the review process.

 Please submit your revised manuscript by Nov 11 2024 11:59PM. If you will need more time than this to complete your revisions, please reply to this message or contact the journal office at plosone@plos.org. Please include the following items when submitting your revised manuscript:A rebuttal letter that responds to each point raised by the academic editor and reviewer(s). You should upload this letter as a separate file labeled 'Response to Reviewers'.A marked-up copy of your manuscript that highlights changes made to the original version. You should upload this as a separate file labeled 'Revised Manuscript with Track Changes'.An unmarked version of your revised paper without tracked changes. You should upload this as a separate file labeled 'Manuscript'.If applicable, we recommend that you deposit your laboratory protocols in protocols.io to enhance the reproducibility of your results. Protocols.io assigns your protocol its own identifier (DOI) so that it can be cited independently in the future. For instructions see: https://journals.plos.org/plosone/s/submission-guidelines#loc-laboratory-protocols. Additionally, PLOS ONE offers an option for publishing peer-reviewed Lab Protocol articles, which describe protocols hosted on protocols.io. Read more information on sharing protocols at https://plos.org/protocols?utm_medium=editorial-email&utm_source=authorletters&utm_campaign=protocols.

We look forward to receiving your revised manuscript.

Kind regards,

Anand Pal

Academic Editor

PLOS ONE

Journal Requirements:

Additional Editor Comments:

Authors have satisfactorily addressed most of the comments of the previous report. I still found lots of grammatical and language issue in the manuscript. I suggest authors once again go through with the whole text of the manuscript.

Reviewers' comments:

Reviewer's Responses to Questions

**Comments to the Author**

1. If the authors have adequately addressed your comments raised in a previous round of review and you feel that this manuscript is now acceptable for publication, you may indicate that here to bypass the “Comments to the Author” section, enter your conflict of interest statement in the “Confidential to Editor” section, and submit your "Accept" recommendation.

Reviewer #1: (No Response)

Reviewer #2: All comments have been addressed

2. Is the manuscript technically sound, and do the data support the conclusions?

Reviewer #1: (No Response)

Reviewer #2: Yes

3. Has the statistical analysis been performed appropriately and rigorously? 

Reviewer #1: (No Response)

Reviewer #2: Yes

4. Have the authors made all data underlying the findings in their manuscript fully available?

Reviewer #1: (No Response)

Reviewer #2: Yes

5. Is the manuscript presented in an intelligible fashion and written in standard English?

Reviewer #1: (No Response)

Reviewer #2: Yes

6. Review Comments to the Author

Reviewer #1: Authors have satisfactorily addressed most of the comments of the previous report. I still found lots of grammatical and language issue in the manuscript. I suggest authors once again go through with the whole text of the manuscript.

Reviewer #2: Based on the submitted revised version of this work, the authors have addressed the requested points. Accept.

7. PLOS authors have the option to publish the peer review history of their article (what does this mean?). If published, this will include your full peer review and any attached files.

Reviewer #1: No

Reviewer #2: No

---

## [Author Response · Author response to Decision Letter 1]

15 Oct 2024

We improved the whole manuscript in terms of Language and uploaded into the author's centers under the name of "Grammatically Improved Manuscript Recent Draft" as requested by the reviewer. Please inform us if there anything else to be done for the publication process.

---

## [Editor Report · Decision Letter 2]

22 Oct 2024

Pressure dependence of the structural and optoelectronic properties of Pb-free perovskites LiSnX3 (X=Br and Cl): A DFT approach.

PONE-D-24-17093R2

Dear Dr. RAMOO,

We’re pleased to inform you that your manuscript has been judged scientifically suitable for publication and will be formally accepted for publication once it meets all outstanding technical requirements.

Kind regards,

Anand Pal

Academic Editor

PLOS ONE
---

## [Editor Report · Acceptance letter]

22 Nov 2024

PONE-D-24-17093R2 

PLOS ONE

Dear Dr. Rammoo, 

I'm pleased to inform you that your manuscript has been deemed suitable for publication in PLOS ONE. Congratulations! Your manuscript is now being handed over to our production team.

Kind regards, 

on behalf of

Dr. Anand Pal 

Academic Editor

PLOS ONE